A-RESS new dynamic and smart system for renewable energy sharing problem

Toumia Imen 1 toumia.imen@gmail.com
Ben Hassine Ahlem 2
1 University of Manouba, National School of Computer Sciences , Manouba , Tunisia
2 University of Jeddah, College of Computer Science and Engineering , Jeddah , Saudi Arabia
Galán José Manuel
Electronic publication date: 2021 Jun 28
Publication date: 2021
Volume: 7
Electronic Location ID: e610
Received 2020 Nov 23; Accepted 2021 Jun 3
Copyright: © 2021 Toumia and Ben Hassine
Copyright year: 2021
Copyright holder: Toumia and Ben Hassine
License: This is an open access article distributed under the terms of the Creative Commons Attribution License, which permits unrestricted use, distribution, reproduction and adaptation in any medium and for any purpose provided that it is properly attributed. For attribution, the original author(s), title, publication source (PeerJ Computer Science) and either DOI or URL of the article must be cited.
License URL: https://creativecommons.org/licenses/by/4.0/

Keywords: Renewable energy sharing, COP, Multi-agent system

Funding: College of Computer Science and Engineering, University of Jeddah, Saudi Arabia UJ-02-078-DR This research was supported by the project UJ-02-078-DR of distinctive research from the College of Computer Science and Engineering, University of Jeddah, Saudi Arabia. The funders had no role in study design, data collection and analysis, decision to publish, or preparation of the manuscript.

==============================
Energy is at the basis of any social or economic development. The fossil energy is the most used energy source in the world due to the cheap building cost of the power plants. In 2017, fossil fuels generated 64.5% of the world electricity. Since, on the one hand, these plants produce large amount of carbon dioxide which drives climate change, and on the other hand, the storage of existing world fossil resources is in continuous decrease, safer and highly available energy sources should be considered. Hence, for human well-being, and for a green environment, these fossil plants should be switched to cleaner ones. Renewable energy resources have begun to be used as alternatives. These resources have many advantages such as sustainability and environmental protection. Nevertheless, they require higher investment costs. In addition, the reliability of many planted systems is poor. In most cases these systems are not sufficient to ensure a continuous demand of energy for all in needy regions because most of their resources are climate dependent. The main contributions of this research are (i) to propose a natural formalisation of the renewable energy distribution problem, based on COP (Constraint Optimisation Problem), that takes into consideration all the constraints related to this problem; (ii) to propose a novel multi-agent dynamic (A-RESS for Agent based Renewable Energy Sharing System) to solve this problem. The proposed system was implemented and the obtained results show its efficiency and performance in terms of produced, consumed and lost energy.

Introduction

Fossil fuels are the most used resources mainly to power homes and cars, i.e., in 2017, fossil fuels generated 64.5% of the world electricity. For a country, it is convenient to use coal, oil or gas energy sources to meet its energy needs, but these fuel sources are often limited. In addition, the intensive use of these energies causes dangerous consequences of the environment, mainly the phenomenon climate change (Dincer, 1999; Goldemberg, 2006). The burning of fossil fuels sends greenhouse gases into the atmosphere, polluting air, soil and water while trapping the heat of the sun and contributing to a global warming. Other traditional energies use biomass which designates the organic waste that can become a source of energy after combustion (wood energies), the anaerobic digestion (bio-gas) and other chemical transformations (bio-fuel). However, a biomass plant operates in a very similar way to fossil power plant. These pollutant fuel plants have serious consequences on the environment and on humans. Hence, even with an unlimited stock of fossil fuels, it is better to use renewable energy for the sake of humans and environment.

Renewable energy is an inexhaustible source of energy because it is constantly renewed by natural processes. Renewable energy is derived from natural phenomena mainly sun (radiation), moon (tide) and earth (geothermal). There is also energy generated from the water and the wind. All these renewable mentioned sources are called new energies. Renewable energy technologies transform these natural sources into several forms of usable energy, most often electricity but also heat, chemical and mechanical energies. Renewable energy technologies are called “green” or “clean” because they pollute little or nothing of the entire environment. The use of renewable energy hybrid power plants allows any country to develop its energy independence and security. Nevertheless, these resources are weather and location dependent, leading to the intermittent and randomness of their use for energy production. The hybrid power plant production fluctuates independently from demand, yielding to an energy excess for some power plants while others cannot satisfy their minimal needs.

Since the production of renewable energy is expensive due to the high cost of both the installation of renewable energy power plants and the storage devices, the optimal solution is to avail from the produced quantity of this energy by maximizing its use and consequently minimizing its loss.

Many attempts have been made in the past decade to enhance, generalise and mainly optimise the use of renewable energy using different technologies, including meta-heuristics (Piccolo & Siano, 2009; Senol et al., 2016; Kumawat et al., 2017; Etxeberria et al., 2010; Niknam & Firouzi, 2009; Soroudi, Ehsan & Zareipour, 2011) for an optimal DGs energy production plan; chance-constrained programming (Li et al., 2018) as a stochastic programming for improving DGs performance; mixed linear programming (Omu, Choudhary & Boies, 2013), peer-to peer platforms and projects (Kusakana, 2020; Shichang et al., 2020a; Shichang et al., 2020b; Kusakana, 2019; Klein, Matos & Allegretti, 2020), Stackelberg game (Xu et al., 2020; Li et al., 2021), machine learning (Amit & Ivana, 2019) for an optimal energy sharing system between a group of buildings; fuzzy multi-objective linear programming (Incekara, 2019) for the optimization of the best energy mix. Most of these efforts (i) deal with sharing energy at a local levels only, while our goal is to consider global levels, for different environment and under several constraints, (ii) focus on how to find the best plan for locating and using distributed generators (DGs), and not how to optimise the profit from the produced energy, (iii) seek for the best renewable energy source or combination of sources to use, and not the best way to coordinate between these sources, (iv) rely on storing energy excess for later use despite its high cost, (v) do not consider the dynamic intrinsic aspect of these resources and (vi) do not consider the advantages of energy sharing among several plants/countries mainly on the environment.

The goal of this work is to propose a novel dynamic, distributed and smart system (A-RESS for Agent based Renewable Energy Sharing System) that (i) maximises the use of produced renewable energy (consequently minimizing its loss) by sharing the excess of energies between countries, and (ii) minimizing the production cost (production, storage and transportation) by investment recovery.

The main contributions of this research are (i) to propose a natural formalisation of the renewable energy distribution problem based on the Constraint Optimisation Problem (COP), that takes into consideration all the constraints related to this problem; (ii) to propose a novel multi-agent dynamic to solve this problem.

The reminder of this paper is organised as follows. “Related Work” describes several research efforts that deal with enhancing the use of renewable energy. A detailed explanation of the energy sharing problem followed by an illustration of our proposed COP formalisation for this problem are given in “New COP Formalization for Renewable Energy Distribution Problem”. A multi-agent dynamic of the proposed solver is given in “A-RESS Global Dynamic”. Finally, a description of the different scenarios of the performed experimentation followed by an explanation of the obtained results are drawn in “Experimental Evaluation”.

Related Work

Up to now, there have been several important pioneering research efforts toward enhancing the use of renewable energy as an alternative solution for mainly electricity and heat demands. Existing researches can be classified into three groups. The first group concerns efforts that consider the problem of locating DGs (Distributed Generators) whether these generators are dedicated to renewable or non-renewable energy. The second group presents works dealing with the planning of Renewable Energy Sources (RES). While the third group concerns the optimization of renewable energy sharing at local level, within a building or among buildings of a same region.

In the first group, Piccolo & Siano (2009) considered the issue of satisfying demands’ growth and distribution network security. They indicated the potential of DGs in offering an alternative approach to utilities compared to centralised generator. As DG provides many benefits for DNO (Distribution Network Operators). The main goal of this work is to evaluate the impacts of network investment deferral, if recognized to DNOs, on DG expansion. The proposed approach allows consideration of variable energy sources in condition to the deterministic ones. As for Senol et al. (2016), they suggested the use of a control method based on Swarm Intelligence (SI). Their goal is to increase the robustness of the 3-phase DMC (Direct Matrix Converters) control system. A DMC is a type of AC-to-AC power converter having relatively small dimensions of circuits which provide substantial advantages for them. The idea consists of generating optimal switching states by using a swarm optimization algorithm and also applying them to the power switches of DMC. As for Kumawat et al. (2017), authors also proposed the use of the meta-heuristic swarm intelligence to find the optimal energy production plan for DGs that minimises energy loss. The particle-swarm-optimization meta-heuristic has been used to determine the optimal size and allocation of the DGs to fulfill consumer demands and reduce power losses. The authors in Li et al. (2018) proposed a two-stage method for determining the optimal locations and sizes of DGs in distributed networks with the integration of energy storage. This method uses the Chance-Constrained Programming to determine the maximum outputs of energy storage devices. They showed that the integration of energy storage is an effective way for DG powers to achieve their pre-designed rated capacity at planning stage and consequently improve their powers’ output performances.

As for the second group of researches, which considers the problem of selecting the best energy source or combination of sources according to their economic and environmental impacts, Niknam & Firouzi (2009) proposed a practical Distribution State Estimation (DSE) system that includes Renewable Energy Sources. Their system is based on the combination of Nelder-Mead simplex search and Particle Swarm Optimization (PSO) algorithm, called PSO-NM, that can estimate load and RES output values by Weighted Least-Square (WLS) approach. Their obtained experimental results compared to other evolutionary optimization based systems such as original PSO, Honey Bee Mating Optimization (HBMO), Neural Networks (NNs), Ant Colony Optimization (ACO), and Genetic Algorithm (GA) demonstrate that PSO-NM is extremely effective and efficient for the DSE problems. As for, Etxeberria et al. (2010), they compared and analyzed several methods using the Hybrid Energy Storage System (HESS) to facilitate the integration of Renewable Energy Sources (RES) into the micro-grid context. The micro-grid is considered as an alternative solution for the problem of stability, reliability and power quality in the main electrical grid. A HESS is usually formed by two complementary storage devices that can be associated in several topology. While Sadegheih (2010) and Sadegheih (2011) presented the use of Mixed Integer Programming (MIP), GA, SA and TS to solve the problem of minimum cost expansion of power transmission network under carbon emission trading programs. Regards AlRashidi & EL-Naggar (2010), they proposed a PSO based algorithm for annual load forecasting in an electrical power system with the aim of minimizing the error associated with the estimated model parameters. Soroudi, Ehsan & Zareipour (2011) proposed an Immune Genetic based Algorithm (I-GA) to present a long-term dynamic multi-objective planning model for distribution network expansion along with distributed energy options. This algorithm optimizes costs and emission by determining the optimal schema of sizing, placement and dynamics of investments on DGs and network reinforcements over the planning period. As for Lasseter (2011), they tried to improve the reliability (through penetration of renewable sources), dynamic island, and generation efficiencies through the use of waste heat. Their objective is to improve the management of the different levels of Distributed Energy Resources (DERs) with the underlying resources and control points by breaking the whole system into micro-grids. Hence, the clustered sources and control loads can operate in parallel to the grid. This grid resource can be disconnected from the utility during events, but may also be intentionally disconnected when the quality of power from the grid falls below certain threshold. The authors in Omu, Choudhary & Boies (2013) studied the economic and environmental impacts of the use of distributed energy generating systems (including renewable energy) compared to centralised one. Therefore, they proposed a Mixed Linear Programming (MLP) model for the design of a system that meets the electricity and heating needs of a cluster of buildings. As for Incekara (2019), they proposed a fuzzy Multi-Objective Linear Programming (MOLP) to obtain Turkey’s energy mix in 2035. They used fuzzy techniques to obtain energy objectives. The authors in Xu et al. (2020) presented a new two-stage game-theoretical framework of residential PV panels planning. First, they use a Stackelberg game theory to model stochastic bi-level energy sharing problems. In this stage, they proposed a descend search algorithm-based solution to obtain an optimal installation capacity of residential PV panels. Then, they developed an Optimal Power Flow (OPF) model to optimally allocate residential PV panels with minimum expected active power loss.The objective of the model proposed in Li et al. (2021) is to maximize the benefits of the Integrated Energy Operator (IEO) which is the difference between the total incomes and the operating cost. To achieve this goal, the authors proposed a Stackelberg game optimization framework for Integrated Energy System (IES) scheduling through coordination of Integrated Demand Response (IDR) and renewable generation. In this game model, the users are the followers who adjust energy consumption to minimize their energy costs and the IEO is the leader who maximizes his profits by setting prices.

For the third researches group, Rafik et al. (2019) proposed a new architecture Micro Smart Grid by Software-Defined Network (MSGSDN). This architecture adopts the Software-Defined Network (SDN) approach and uses the Internet of Things (IoT) sensors to allow an efficient and intelligent distribution of electrical energy obtained from several sources within many buildings. The MSGSDN architecture deals with several constraints: the preference of energy sources, the type of the building devices, the real time of equipment’s consumption and its security. As for Aldaouab, Daniels & Ordonez (2019), they tried to optimize the energy exchange between two prosumers. Their optimisation problem is defined by a Model Predictive Control (MPC) framework based on future behaviour prediction algorithms. The two used prosumers have the same structure: loads, energy supply, battery storage and connections to other power sources. The power flows transfer is done only between two prosumers through the peer-to-peer transactions block. Azizi et al. (2019) proposed an autonomous and decentralized power sharing and energy management approach for PV and battery based DC microgrids without utilizing a supervisory and communication system. As for Raffaele & Mariagrazia (2019), they presented a decentralized control strategy for the scheduling of electrical energy activities of microgrids. The microgrid is composed of smart homes connected to a distributor and exchanging renewable energy produced by individually owned distributed energy sources. The authors assume that each smart home can both buy/sale energy from/to the grid taking into account time varying non linear pricing signals. Amit & Ivana (2019) modeled the work environment as a multi-agent environment where each agent represents a building and they proposed a Deep Reinforcement Learning (DRL) solution to optimize energy sharing between different buildings. The intelligent agent learns the suitable behaviours to share energy in order to realize a nearly zero energy status. Kusakana (2020) proposed a peer-to-peer energy sharing model. The advantage of their model is to minimise the operating cost of the two prosumers by maximising the use of the power from renewable energy sources and minimizing the use of the electrical supplying energy under the time of use rates. Akter et al. (2020) proposed a distributed approach to solve the problem of optimizing energy management between different houses (house without solar photovoltaic or battery, with solar photovoltaic as well as batteries). The proposed approach allows different houses to make decisions without sharing any information with the central transactive energy management system. The proposed approach is efficient in terms of optimal use resources and efficient sharing of energy between different houses in microgrids. As for Shichang et al. (2020a), they presented a peer-to-peer energy sharing framework for numerous community prosumers. Two strategy are proposed in this research, intercommunity energy-sharing and an intracommunity energy-sharing for day-ahead and relative energy-sharing schedule respectively. Shichang et al. (2020b) suggested a new and fair peer-to-peer for a community of energy buildings. First, a generalized Nash equilibrium (GNE) of the game is displayed independently of energy sharing payments. Then a cost reduction ratio distribution (CRRD) model is used to fix energy sharing payments for the buildings. Klein, Matos & Allegretti (2020) proposed an end-user engagement framework tailored to fit the peer-to-peer (P2P) energy sharing. The objective of the proposed P2P energy sharing model is to optimize the energy consumption of each network participant according to the available energy distributed. The network in this study is made up of a static number of pilots (set to three) and the communication established between them is static. While Andrea et al. (2021) presented a two-stage approach that allows sharing renewable energy within the district by using a set of prosumers and aggregators that supervises the energy exchanges between the prosumers and with the grid. The use of the two-stage approach minimizes the revenues deriving from the prevision and the sale of energy. As for Kusakana (2019), they developed an optimal energy management model between commercial prosumers and a residential consumers. This model allows to minimize the cost of energy consumed. The proposed model has been evaluated in a microgrid using peer-to-peer energy sharing schemes operating Time of Use Tariff.

All these efforts seek the optimization of the planning and use of energy (including renewable one) using different techniques. Nevertheless, most of their main goal is to find the best location (using meta-heuristics) of their DGs according to demands and load clusters, meaning to decide whether or not to establish a renewable energy system in a given place and which renewable energy system source or combination of sources is the best choice. Planning of the energy system is to select the best alternative among the different renewable energy systems (Baños et al., 2011). Other researches goal was devoted toward finding solutions for storing the extra-clean obtained energy for later use despite the cost and underlying energy loss. Researches, that treats the energy sharing problems, studied the case of energy sharing in a limited and static space, between buildings belonging to the same region or within the same building. Note that, compared to our A-RESS system, none of these efforts dealt with establishing a dynamic and naturally distributed energy sharing system that (i) considers all constraints related to this problem mainly physical distances, (ii) aims to maximise local and global satisfaction of several plants/countries subject to production fluctuation and depending on weather conditions that may lead a supplier to become a consumer and (iii) minimises local and global energy losses. Recall that our main objective is to optimise the use of clean energy between several countries, mainly those in lack of renewable resources for a green environment.

New COP Formalization for Renewable Energy Distribution Problem

Description of the problem

Depending on the geographical location of the regions, renewable energy sources may change, e.g., some of the countries are sunny allover the year, while for others the sun comes out for only few days. Hence, a country with a continuous and important opportunity of renewable energy production is known as a producing country, while a country which cannot meet its proper needs is known as a consuming country. Since, the production and storage of renewable energy are expensive and in order to optimise these costs while ensuring a clean environment, a system with a local and global regulator for energy sharing can optimise the life cost of the renewable energy powers and reduce their loss for a green environment.

Our aim is to build an undirected and dynamic graph connecting the different N hybrid power plants (of different involved countries). The graph is dynamic, since a power plant can be added/deleted at any time and it is undirected because any power plant can switch from an energy supplier to an energy consumer and vice versa. The graph is incomplete because it is useless to connect power plants which are at large distances. Each hybrid power plant Xi, i ∈ {1,…, N} in a region Ri, maintains important information as:The estimated quantity of renewable energy that can be produced Prod(Xi) per day,

The remaining quantity of renewable energy in batteries Stock(Xi) per day,

The estimated quantity of needed energy Needs(Xi) for the region under several conditions, i.e., current season, temperature, etc.

The quantity of energy to keep in reserve Reserve(Xi) for any risk.

For predicting these data, i.e., Prod(Xi) and Needs(Xi), and for a high performance and accuracy for the proposed A-RESS system, several existing models can be adopted for estimating energy production and consumption. Most of them are based on Artificial Intelligent and machine learning techniques, amongst, artificial neural network (ANN) based model that have provided good results for real-time prediction of energy production (Bermejo et al., 2019; Shapi, Ramli & Awalin, 2021); Multiple Linear Regression (Solyali, 2020); Support Vector Machine (Kaytez, 2020; Walker et al., 2020); Random Forest (Walker et al., 2020); K-Nearest Neighbour (Shapi, Ramli & Awalin, 2021); Multilayer Perceptron (Chammas, Makhoul & Demerjian, 2019); etc. The prediction of the required quantities of energy is based mainly on daily meteorological data and production/consumption histories for a given period. The latter is divided into two types of features: Historical features, i.e., energy previous consumption, and weather features, i.e., temperature, humidity, solar radiation, wind speed, wind direction, pressure, rainfall amount, degree of cloudiness, type of day (weekday/weekend/holiday), type of hour (daytime/night-time), season, etc. Most studies cited above showed the effectiveness of Artificial Neural Network (ANN) and Support Vector Machine (SVM) based models. The integration of an ANN or an SVM model in our system for the required daily data will be considered in the future work.

Once the daily quantity of energy to provide by each supplier Xi to its consumer Xj neighbors is determined (using Eq. 3). The transfer of renewable energy can be done at a local level means between plants of the same country; in this case the energy will be offered. The energy can also be sold when the transfer involves plants of different countries. In this work, we will focus on the international scale. For simplifying the proposed system, we assume that each country contains one single power plant Xi. A transfer cable is provided between two power plants Xi and Xj only if the distance between these two plants is not large. Hence Xi and Xj are considered as neighbors ,i.e., Ng(Xi). Our goal is to optimise the shared quantity of renewable energy while minimizing the cost of transportation and maintaining a high level of transportation reliability. The possessing of the transmission line depends on the quantity of energy to be transported and the distance between the two power plants to interconnect. Note that each transfer process will obviously lead to some loss of energy.

The decision of any power plant for whether acquiring some quantity of renewable energy or producing the same needed quantity of non-renewable energy is based on the basic price of the plant’s installation, the amortization period, the transfer cable installation, etc. In addition, any Xi can deliver or store energy if and only if its both needs and risk reserve are satisfied. For reducing the loss of energy, the quantity received by Xi must not exceed its needs/risk reserve.

New formalization for renewable energy sharing problem

A renewable energy distribution problem can be formalised as a Constraint Optimisation Problem (COP). A COP is a Tuple(X, D, C, F) as follows:X = {X1, X2,…, Xn} with Xi = (Xi.io1, Xi.io2,…, Xi.iok), where Xi represents a power plant and Xi.ioj, j ∈ {1,…, ||Ng(Xi)||}, is the quantity of energy to provide/get to/from Xj, j ≠ i. Note that Xj ∈ Ng(Xi) only if these two plants are linked together, i.e., Ng(Xi) is the set of all neighbors of Xi,

D = {D1, D2,…, Dn} where Di = (val1, val2,…, valk) with valj ∈ N, are the possible quantities of renewable energy to provide/get to/from each Xj ∈ Ng(Xi). If Xi is a power plant provider then ∀ Xj∈ Ng (Xi), Xi.ioj ≥ 0.

C = {CProf, CShare, CRec} where

– Profitability Constraint CProf: the cost of requesting some quantities of renewable energy from neighbors should be better than producing the same quantity of non-renewable energy (see Eq. 1). These costs involve: the cost of the transfer, the cost of the cable installation, the cost of air pollution, etc.

(1) ∑j∈Ng(Xi)(βij∗Xi.ioj)−(γi∗∑j∈Ng((Xi)(1−αij)∗Xi.ioj)<0

where

αij is the proportion of lost energy, determined according to the link type between Xi and Xj, its length, the weather condition, etc. γi is the cost of producing 1 KW of non-renewable energy by the plant Xi, and βij is the cost of exchanging 1 KW of renewable energy between two plants Xi and Xj.– Shared Quantity Constraint CShare: the total shared quantity per plant should not exceed the available quantity of energy to be given as in Eq. (2).

(2) ∑Xj∈Ng(Xi)Xi.ioj≤Avail(Xi)

where

(3) Avail(Xi)=Prod(Xi)+Stock(Xi)−Needs(Xi)−Reserve(Xi)

– Received Quantity Constraint CRec: the total received quantity of renewable energy per a plant should not exceed the needed energy as in Eq. (4).

(4) ∑Xi∈Ng(Xj)(1−αij)Xj.ioi+Avail(Xj)≃0

Solving this COP consists in finding the optimal (according to the objective function F) instantiation of all variables X with values of their domains D that satisfies all the constraints C. An optimal solution X * = {X*1, X*2, …, X*n} for this problem is a solution that maximizes the satisfaction of all neighbors requests, F(X *). Each power plant tries to compensate the spent cost of its installation by serving the maximum in needy regions with possible quantities. In addition, this system tries to maximize the benefit from distributing energy. The cost of purchasing/selling energy differs from one plant to another according to the price of the link installation, the distance between the regions, etc.

Each plant Xi will try to find values X *i = (X*i.io1,X*i.io2,…, X*i.iok) with k = ||Ng(Xi)||, that optimizes F(X), as shown in Eq. (5).

(5) F(X∗)=minX⁡F(X)

(6) F(X)=g(X)−h(X)

where, g(X), is the function that measures the degree of non-satisfaction of all the plants Xi ∈ X (as given in Eq. 7). An energy plant supplier Xi is satisfied only if the maximum of its extra produced energy is shared with neighbors, while an energy plant consumer Xj is satisfied only if it gets the maximum of the energy it needs. As for h(X), it computes the global cost of all the energy exchanged between all the plants (as given in Eq. 8). If h(X) > 0 then most plants are suppliers; otherwise most plants are consumers.

(7) g(X)=∑Xi∈X(∑Xj∈Ng(Xi)Xi.ioj−Avail(Xi))2

(8) h(X)=∑Xi∈X∑Xj∈Ng(Xi)βij∗Xi.ioj

Recall that βij is the cost of exchanging 1 KW of energy between two plants Xi and Xj. This cost is not the same for all plants, it is subject to type and length of the used cable, the installation cost of the plant, etc. Both quantities, h(X) and g(X) should be normalized.

A-RESS Global Dynamic

Since our problem is naturally distributed where each plant is responsible for taking its own decisions upon its knowledge, we propose to use a multi-agent system for solving this problem. Agents will communicate to agree on quantities to exchange, that maximises the satisfaction of them all, i.e., F(X*).

System architecture

In the multi-agent proposed system, each agent will be assigned to a power plant. Three types of agent are used, a Supplier, a Consumer and a Neutral Agents. Supplier agents are those who have an excess of renewable energy and subsequently can provide some of it to the neighbors who are indigent. These latter agents are the Consumer. A neutral agent is an agent that is able to satisfy its needs and does not have any extra energy.

Each agent Xi maintains five static knowledge (i) Estimated quantity of renewable energy that its power plant can produce by per day, (ii) Available quantity of renewable energy in its power plant batteries per day, (iii) Estimated quantity of needed energy for its power plant, (iv) Quantity of energy to keep in reserve for any risk and (v) Number of all its neighbors ||Ng(Xi)||. Agents also have dynamic knowledge, this knowledge differs according to the type of agent. A supplier might be mapped into a consumer if it faces a lack of energy at any day, same for a Consumer. Agents can be mapped according to their daily needs. All these types of agents have to cooperate together in order to maximize their own satisfaction.

Global dynamic system

The proposed multi-agent global dynamic is divided into two phases:An initialization phase

A negotiation phase

During the initialization phase, each agent will use its trained ANN model to estimate the production and the needs of the day. As we mentioned before this process will be discussed in another research work. According to the obtained estimations, decides the quantity of available renewable energy to share with neighbors. If the Avail(Xi) is a positive value then Xi will be a Supplier; otherwise it will be a Consumer. Agents can then start the second phase.

During the negotiation phase, each Consumer agent, will check first whether is it profitable to get the needed quantity or to use non-renewable energy according to Eq. (1). If yes, then it will send a request to all its neighboring Supplier, with the needed quantity i.e., the agent will divide the needed quantity |Avail(Xi)| on the number of Suppliers. Each Supplier Xj that receives demands from related Consumers, proceeds first by checking whether it is possible to grant them all, i.e., Avail(Xj) is equal or greater than the summation of all requested quantities. If so, each Consumer will receive what it needs and the extra-quantity will be kept for further requests. Otherwise, the agent Xj will determine the proportion pij of each request according to the total needs (summation of all asked quantities from all Xi ∈ Ng(Xi)). Xi will reply to all its requesters with possible quantities (see Eq. 9).

(9) Xi.ioj=pij∗Avail(Xi)

(10) pij=Xi.ioj∑Xj∈Ng(Xi)Xi.ioj

Each Consumer agent Xi will check the total amount of energy that it is willing to get from all Suppliers. If the whole quantity is equal to its need (satisfying constraint of the Eq. 4) and optimise its objective function F(Xi), it replies with an acceptance. Otherwise, if it is less than its needs (due to some energy loss or non availability of enough energy on the Supplier side), then it will resend to other Suppliers, that are willing to provide more, asking them for the missing quantity. The same negotiation proceeds until all available quantities are distributed and all Consumers get the maximum of what they can get. Once the negotiation process is over, the power plant may start exchanging their agreed quantities.

System architecture description

In this section, we will describe our multi-agent system using the ODD (Overview, Design, and Details) protocol proposed by Grimm et al. (2006).

Overview

1. Purpose: The objective of our multi-agent model is to specify the quantities of energy to be released from agents with energy excess to agents with a lack of energy to satisfy their needs.

2. Entity, states variables and scale: Our model is made up of three types of agents: (1) Supplier Agent: represents power plants that produce a quantity of energy that exceeds their needs, (2) Consumer Agent: represents power plants that produce a quantity of energy that does not satisfy their needs and (3) Neutral Agent: are the power plants that produce a quantity of energy that only meets their proper needs.

Each agent knows the quantity of energy that has in its storage devices (Stock(Xi)), the quantity of energy to keep in reserve for all risks (Reserve(Xi)) and the number of its neighbours (||Ng(Xi)||).

The agents are located in an environment made up of a set of regions where each region has meteorological characteristics (temperature, humidity, amount of rain, cloudiness, season, etc.) and a machine learning model to predict its daily quantity of energy (4) to be produced (Prod(Xi)) and (5) to be consumed (Needs(Xi)).3. Process and scheduling: Every 24 h, the following processes are executed in this given order:

Each Consumer agent sends the quantity it needs to each of its Supplier neighbours.

Supplier agents receive all requests from neighbouring Consumer agents.

Each Supplier agent compares the total quantity requested with the quantity available. If the total quantity can be provided, then it responds to each Consumer with the requested quantity. Otherwise, it determines the quantity to be given. The quantity Xi.ioj (Eq. 9) to be offered from the Supplier agent Xi to a Consumer agent Xj is determined according to availability Avail(Xi) (Eq. 3) and the portion pij given by the equation Eq. (10). This proportion is assigned to each consumer depending on its environment, i.e., strong consumer with many suppliers, or weak consumer with few suppliers.

Each Consumer agent receives the responses from all the neighbouring supplier agents and checks the total of the quantities received with its need. If the total is less or equal to its needs, it accepts the proposed quantities, otherwise it adjusts the quantities according to its needs and sends a response with only needed quantities.

Design concepts

Emergency: Each agent has a local goal that is to provide all its energy excess. This goal is represented by the function F(X *) given by the equation Eq. (5), which aims to minimize the two functions g(X) and h(X) (see Eqs. (7) and (8)).

Sensing: For a Supplier agent, in all cases the total quantity of the shared energy does not exceed its available quantity of renewable energy. The same for an agent consumer, i.e., the total quantity of received energy does not exceed its needs under any circumstance.

Interaction: The only interactions are among each Consumer and its neighboring Suppliers. Neutral agent does not interact with other agents because it is already satisfied, i.e., it has neither lack nor excess of energy. Supplier agents interact with Consumer agents to negotiate the quantities of energy to be shared. These interactions resume until no more available energy to be shared.

Stochasticity: In the initialization phase, there are two types of data: known data and data to be estimated. The quantity of energy stored in the battery (Stock(Xi)) and the energy reserve (Reserve(Xi)) of each power plant are both known quantities. However, the quantity of energy to be produced (Prod(Xi)) and the quantity of energy to be consumed (Needs(Xi)) are predicted using a trained regression model.

Details

Initialisation: The first action to be performed is the estimate the quantity of energy to be produced (Prod(Xi)) and the quantity of energy to be consumed (Needs(Xi)) for each agent. Then, each agent calculates its quantity of available energy (Avail(Xi)). If the quantity is positive, then it is a Supplier agent otherwise it is a Consumer agent. The agents exchange their states then each Consumer agent calculates the number of its Supplier neighbors. Based on the number of neighbors and the total number of agents, Consumer agents are classified into two types: Strong Consumer who have more neighbors and those who have fewer are Weak Consumers.

Input Data: The model takes as input a graph containing the list of power plants and the links between them. Two power plants are linked, meaning that they are neighbours and can interact.

System complexity

In order to evaluate the computational efficiency of the proposed system and decide whether it can be used in practical applications or not, we computed the time complexity of the underlying multi-agent system. Since a multi-agent system is defined as a set of interacting agents in an environment, we need to determine the complexity at two levels, the agent level (as an atomic unit) and the system level (including the interactions). In the following, we assumed N the total number of agents, each agent is connected at most to N/2 other agents as neighbors.At the agent level: According to its behavior, each agent has to predict the quantities of renewable energy to be produced and to be consumed in order to estimate the quantity to be offered. The complexity of this process depends on the trained regression model that will be used. Note that most of the existing models are polynomial. Then the agent has to process m(N/2) values to/from neighbors, with m the total number of iterations to attend consumers’ satisfaction. In the worst case, only one Consumer will be satisfied at each communication, so the total number of iterations for a Supplier will be N/2. Hence, each consumer will process O(N2) instructions.

At the system level: According to the designed interactions, a first communication is required between agents to exchange their states (Consumer or Supplier). This first step requires O(N2) messages at most. Then, Consumer agents will negotiate with their Suppliers to get what they can get. These communications sending/receiving requests/answers between agents resumes until all consumers are satisfied (no more available quantities to be shared). Assume that at each communication one neighbor is satisfied, so the total communications will be equal to the number of neighbors for each agent, means O(N2). All agents will process O(N3) messages.

Illustration example

To illustrate the global dynamic of our proposed system, let’s consider the following example: assume we have five regions {X1, X2,…, X5} described in the Table 1.

Table 1 Illustration example.

Distance	X1	X2	X3	X4	X5	
X1	0	d12	d13	d14	d15	
X2	d12	0	d23	d24	d25	
X3	d13	d23	0	d34	d35	
X4	d14	d24	d34	0	d45	
X5	d15	d25	d35	d45	0	

To represent the distance graph, we assume that the two regions Xi and Xj are linked if the distance dij is less than the maximum distance D. In our example, we assume that d15, d13, d23, d24, d45 ≥ D as shown in the following graph:

The Table 2 gives an example of the estimated energy per region.

Table 2 Production example.

	X1	X2	X3	X4	X5	
EstProd	6,000	8,000	8,000	4,000	11,000	
QteStock	1,000	1,000	2,000	3,000	500	
EstNeed	7,000	500	7,000	8,000	6,000	
Reserve	2,000	3,000	3,000	500	4,000	
Avail(Xi)	Consumer (−2,000)	Supplier (+1,000)	Neutral	Consumer (−1,500)	Supplier (+1,500)	

During the initial phase, agents will first estimate their energy available quantity and decide about their current status, i.e., whether a Consumer, a Supplier or a Neutral agent. Then they will exchange their status as given in the following graph:

During negotiation phase, the Consumer agents will send a message to all neighboring Suppliers to ask for the needed quantities. X1 will ask for 2,000 KW from X5 and X4 will ask for 750 KW from X2 and X5. Then X5 will compute p51 = 0.72 and p54 = 0.28, and inform X1 and X4 about the quantities they are willing to receive, i.e., 1,080 KW for X1 and 420 KW for X4. As for X2, it will inform X4 that it is able to get only 500 KW. The Consumer agent will accept the offer given by Suppliers, since no additional affordable energies as given in Table 3.

Table 3 Negotiation phase results.

	X1	X2	X3	X4	X5	
Avail(Xi)	−2,000	+1,000	0	−1,500	+1,500	
Supplier/Quantity	X5/750	–	–	X2/500	–	
				X5/750		
NEW Avail(Xi)	−1250	0	0	−250	0	

Experimental evaluation

For an effective performance evaluation of our A-RESS system, we need to show that: (i) All produced quantities of renewable energy will be shared among involved power plants, and (ii) None of the power plants that lack energy, gets more than its demand. This system can be adopted by one large country, e.g., Saudi-Arabia or by several neighboring countries. The aim is to guarantee both, the maximum use of produced renewable energy for a less environment damage and the minimum loss of energy. We implemented a prototype of our system using JADE environment, a Java programming language on an Intel (R) Pentium (R) Dual CPU T3200 processor with a Microsoft Windows 7 operating system.

Several experiments were performed under the following assumptions:The number of countries is 10, {X1, X2,…, X10} described in the Table 4.

For each country only one power plant is installed. We will have one agent per country.

A link exists between two countries if the distance between them is less than some threshold, for our experiments is 2,500 km.

A country Xi can be Neutral, Supplier or Consumer according to its maintained data. These data, i.e., Prod(Xi), Needs(Xi), Stock(Xi) and Reserve(Xi), are generated randomly.

Table 4 Distance between countries.

Distance (Km)	Tunisia (X1)	Saudi Arabia (X2)	Bahrain (X3)	Qatar (X4)	Kuwait (X5)	E.A.U (X6)	Yamen (X7)	Oman (X8)	Iran (X9)	Egypt (X10)	
Tunisia (X1)	0	3,614	4,018	4,017	3,609	4,441	4,396	4,726	4,081	2,181	
Saudi Arabia (X2)	3,614	0	3,603	638	649	894	1,458	1,143	1,269	1,470	
Bahrain (X3)	4,018	3,603	0	101	471	443	1,188	744	1,586	1,967	
Qatar (X4)	4,017	638	101	0	572	345	1,124	645	1,903	2,040	
Kuwait (X5)	3,609	649	471	572	0	911	1,534	1,212	686	1,685	
E.A.U (X6)	4,441	894	443	345	911	0	1,038	301	1,001	2,434	
Yamen (X7)	4,396	1,450	1,188	1,124	1,534	1,038	0	1,024	1,947	2,219	
Oman (X8)	4,726	1,143	744	645	1,212	301	1,024	0	1,234	2,817	
Iran (X9)	4,081	1,269	1,586	1,903	686	1,001	1,947	1,234	0	1,981	
Egypt (X10)	2,181	1,470	1,967	2,040	1,685	2,484	2,219	2,817	1,981	0	

The performance of our A-RESS system is evaluated in terms of three metrics: (i) the CPU time for scalability testing, (ii) the energy availability per region before/after sharing and (iii) the difference between afforded/received for sensibility testing. The obtained results are represented in the following Figures.

To determine the selling price of 1 KW of renewable energy, we needed to consider the initial renewable energy power plant and link installation cost β0i, the amortization period ti and the discount rate σi. The selling price of a 1 KW of non-renewable energy is represented by a geometric suite (as given in Eq. 11).

(11) βij=σinβij0+a

with n ∈ [0..ti] and a is a fixed fees.

In our experiments, 30 samples were generated. These samples differ in terms of the estimated quantities of energy consumption, production, remaining quantity in batteries and the quantity to keep in reserve. So, the numbers of Neutral, Supplier and Consumer agents vary from one sample to another.

In Fig. 1, the blue curve shows the total quantities of available energy, while the red curve shows the total quantities of shared energy for all Supplier agents per sample. Note that for most samples both curves superimposed. Only for few cases the available quantity is slightly greater the shared one. This means that most produced energy by suppliers will be used by consumers and consequently the loss of energy is very few.

Figure 1 Difference between available and shared energy for supplier agents.

In Fig. 2, the blue curve describes the total quantities of requested energy and total quantities of received energy for all consumer agents per sample. Notice that the two curves overlap for many samples. This means that for these samples the consumers are well satisfied and they got what they requested. For sample 21, the consumers received only 25% of their requested quantities. This gap between requested/received quantity can be justified but the results given in Fig. 3 where it is shown that for Sample 21, the number of Consumers are far greater than the number of suppliers. In addition, it is worthy to notice that none of the consumers received more than what it requested.

Figure 2 Difference between needed/received quantity of energy for all consumer agents.

Figure 3 Number of supplier agents vs number of consumer agents.

To approve the consistency of the obtained results, Fig. 4 matches the total quantities of energy shared by all the Supplier agents with the total quantities of energy received by all Consumers agents, per sample. Note that the two curves are almost fully superimposed. We infer that all shared quantities have been well received by the Consumer agents.

Figure 4 Total quantities of shared energy vs total quantities of received energy.

From all above obtained results, we believe that despite the variation of the random inputs (Prod(Xi), Needs(Xi), Stock(Xi) and Reserve(Xi)), the proposed multi-agent dynamic is accurate. In all cases, a Supplier agent cannot offer more than what it is available for it, a Consumer agent cannot receive more than it needs and mainly most of the produced quantity of renewable energy is well distributed.

To test the scalability of our approach, we varied the number of plants from 10 to 200 and we measured the required CPU time. Figure 5 shows that the CPU time increases, with polynomial growth, according the number of plants. This is due to the number of messages that are exchanged in order to attend a compromise between all agents. This number is a polynomial function of the number of plants. For 10 power plants we needed 80 s while for 200 power plants, we had 620 s.

Figure 5 Number power plants/CPU time.

Conclusion

In this paper we have described different existing systems for enhancing the use of renewable energy. Most of these efforts consider the problem of locating renewable or non-renewable energy DGs (Distributed Generators) based on meta-heuristics. Others considered this problem as a planning Renewable Energy Sources (RES) problem. While remaining efforts are concerned with the optimization of renewable energy shared at local level. However, none of these efforts deal with sharing these quantities with other countries to satisfy their needs and to improve environmental impact. Our work deals with finding the best way to share extra produced renewable energy with neighbouring countries to improve climate conditions. Therefore, we have proposed a novel smart and agent-based system that is able to share most of the available quantities of renewable energy with all neighboring regions. The obtained results have shown the performance of the proposed agent-based system in terms of accuracy, satisfactions and energy loss. In future work we will, integrate a regression model in our system for predicting all daily unknown energy quantities to be produced/to be consumed.

Supplemental Information

Supplemental Information 1 Java code using JADE to evaluate the performance of our A-RESS system and the example discussed in our article.

The "RESS" folder contains the Java source code. The “Test” Excel file contains the data generated and discussed in our article.

Click here for additional data file.

We acknowledge with thanks the University of Jeddah technical support.

Abbreviations

A-RESS Agent based Renewable Energy Sharing System

ACO Ant Colony Optimization

ANN Artificial Neural Network

COP Constraint Optimisation Problem

CRRD Cost Reduction Ratio Distribution

DER Distributed Energy Resources

DG Distributed Generator

DMC Direct Matrix Converters

DNO Distributed Network Operators

DRL Deep Reinforcement Learning

DSE Distributed State Estimation

GA Genetic Algorithm

GNE Generalized Nash Equilibrium

HBMO Honey Bee Mating Optimization

HESS Hybrid Energy Storage System

I-GA Immune Genetic based Algorithm

IDR Integrated Demand Response

IEO Integrated Energy Operator

IES Integrated Energy System

IoT Internet of Things

KW Kilo Watt

MIP Mixed Integer Programming

MLP Mixed Linear Programming

MOLP Multi-Objective Linear Programming

MPC Model Predictive Control

MSGSDN Micro Smart Grid by Software-Defined Network

NN Neural Network

OPF Optimal Power Flow

ODD Overview, Design, and Details

P2P Peer-to-Peer

PSO Particle Swarm Optimization

PSO-NM Particle Swarm Optimization-Nelder Mead

PV Photovoltaic

RES Renewable Energy Sources

SA Stand Alone

SDN Software-Defined Network

SI Swarm Intelligence

SVM Support Vector Machine

WLS Weighted Least Square

Parameters

αij Proportion of lost energy

βij cost of exchanging 1 KW of Renewable Energy between two power plants Xi and Xj

γi Cost of 1 KW of non renewable energy

σi Discount rate

Avail(Xi) Quantity of energy available

F(X) Objective function

g(X) Function that measure the degree of non satisfaction of all the plants Xi

h(X) Function that compute global cost of all the energy exchanged

N Number of hybrid power plants

Needs(Xi) Estimated quantity of needed energy for the region Ri

Ng (Xi) Set oh neighbours of Xi

pij Portion of each request according to the total needs

Prod (Xi) Estimated quantity of renewable energy that can be produced Xi per day

Ri The region where the power plant Xi is installed

Reserve(Xi) Quantity of energy to keep in reserve for any risk

Stock(Xi) Remaining quantity of renewable energy in batteries per day

ti Amortization time

X Set of variables

Xi Hybrid power plant

Xi.ioj Quantity of energy to provide/get to/from Xi

a Fixed fees

C Set of constraints

D Set of constraints

Additional Information and Declarations

Competing Interests

Author Contributions

Data Availability

The authors declare that they have no competing interests.

Imen Toumia conceived and designed the experiments, performed the experiments, analyzed the data, performed the computation work, prepared figures and/or tables, authored or reviewed drafts of the paper, and approved the final draft.

Ahlem Ben Hassine performed the computation work, authored or reviewed drafts of the paper, and approved the final draft.

The following information was supplied regarding data availability:

The data and code are available in the Supplemental File.

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
