# Peer review of "A-RESS new dynamic and smart system for renewable energy sharing problem"

_PeerJ Computer Science, doi:10.7717/peerj-cs.610_

## Round 0.1 · original submission · Major Revisions

Although both reviewers consider the paper with merit, there are many suggestions that I also think are very relevant before considering the paper for publication.

Reviewer 1 ·

Basic reporting

(1) The presentation is clear and professional English is acceptable.
(2) The references are incomplete and somewhat outdated. A more comprehensive and timely literature survey is desired.
(3) The article structure is acceptable.

Experimental design

1 The topic of this paper fall in the Aims and Scope of the journal.
2 The knowledge gap is not clear.

Validity of the findings

The validity of the findings must be strengthened. More jobs should be done to verify the work.

Additional comments

This paper presents a-RESS new dynamic and smart system for renewable energy sharing problem. The simulation results are also provided. Here, there are some concerns of this reviewer:

Point-by-point list of major recommendations:
1 Author should highlight the research gaps and contribution of the proposed work by comparing the state of the art methods and recent studies.
2 In the introduction section, the literature review must be strengthened. The references is incomplete and somewhat outdated. A more comprehensive and timely literature survey is desired.
3 How scalable is the proposed approach?
4 The theoretical depth of this paper needs to be strengthened.
5 The proposed method might be sensitive to the values of its main controlling parameter. How did you tune the parameters? Please elaborate on that.
6 The computational cost of the proposed approach isn’t discussed in this work. The approach should be computationally efficient to be used in practical applications.
7 The novelty and contribution of the presented work need further justification. Authors need to add more results to thoroughly support the main findings.
8 Authors have not presented limitations of this work. How this work can be extended in future? Although authors have provided various comparative results, however, more details on how proposed phenomenon performs better results against baseline is still missing in the paper.
9 Please specify details of the computing platform and programming language used in this study.

Point-by-point list of minor recommendations:
10 The nomenclature should be included to help the reader to follow the paper conveniently.
11 The italic types of symbols in equations and the main text should be in the same expression.
12 Please improve the quality of figures completely to improve the readability of this paper.

Reviewer 2 ·

Basic reporting

This article covers an important topic, however I have some considerable concerns:
1. The introduction of the article and research is poorly written, uses too many acronyms, does not define concepts, uses hardly any references, and also does not describe the justification of the research, nor does it clearly describe what the article is about. This needs to be updated.
2. The methodology and description of the approach and its application is better, but there are still many major questions. For example, how does the temporal dimension feature? All the demands and supplies should be a function of time, yet this is not clarified anywhere. What is the time step? This is important, especially for renewable energy given the limited predictability a long way into the future.
3. An Agent-Based Model should be described in some detail, such as by the ODD method, but this is not described here. Therefore, the method is not replicable in my mind unless this is fully described.
4. What data is being used as inputs?

Experimental design

The experimental design has not been clearly described, and I can't find information about important aspects such as sensitivity analysis

Validity of the findings

It is unclear exactly what the findings are, so I can't judge the validity.

Additional comments

Given that on the surface your method looks very useful, I suggest more effort is given to describing it in greater detail so that the validity and details can be evaluated adequately.

---

## Round 0.2 · Major Revisions

Although both reviewers think that the paper has done a very important step forward, there are still some important elements (see especially comments of reviewer 2) that can improve your manuscript. We are looking forward to seeing your new version.

Reviewer 1 ·

Basic reporting

no comment

Experimental design

no comment

Validity of the findings

no comment

Additional comments

I would like to thank the authors for this revised version since most of my concerns have been addressed. The paper deserves to be published after minor revisions:
1. Some references lack information such as volume number and page number. Please check and modified the references.
2 The current literature and technique review is still inadequate, and some important works are missing. Please refer to the following studies: "A pragmatic approach towards end-user engagement in the context of peer-to-peer energy sharing", "Optimal distributed generation planning in active distribution networks considering integration of energy storage", "Optimal scheduling of integrated demand response-enabled integrated energy systems with uncertain renewable generations: a Stackelberg game approach".
3 Although the manuscript is well written in terms of English, there are some (very few, indeed) grammatical errors. It is suggested to proofread the paper.

Reviewer 2 ·

Basic reporting

The manuscript has been much improved and the language is clear and mostly unambiguous with good English throughout.

The paper is however severely lacking in references, especially in some sections. For example the Introduction is almost entirely lacking in references. This is critically important to address because the premise of the model needs to be established based on a real need. The methodology is now much better described. I am however missing further discussion about data. The data requirements are unclear and not fully specified, and unless this data can be found, the model will be useless.

The Related work section is a major improvement, but it needs to more clearly specify how this paper and model contributes something new. In clear and succinct language.

The ODD provides much needed added details and this is great.

The structure is now however much improved.

Experimental design

There is no clear research question or hypothesis stated.

Data is not adequately described.

Validity of the findings

I can't quite see how the experimental evaluation is clearly showing the usefulness of the approach. Who will use the model? Is this example representative of real planning examples? I do not consider that the experiment validates the hypothesis. What have you actually proven? It's not clear to me.

Additional comments

Again, the paper has some significant shortcomings that need to be addressed, but the article is now much improved.

---

## Round 0.3 · accepted · Accept

Both reviewers consider that the paper is suitable for publication in the present form.

Reviewer 1 ·

Basic reporting

The paper is written in clear, unambiguous, and professional English;
the literature references are sufficient;
the article structure is well organized,
Figures and tables are acceptable;
the manuscript includes all results relevant to the hypothesis;
the results include clear definitions of all terms and theorems, and detailed proofs.

Experimental design

This research is original primary research within Aims and Scope of the journal;
the research gap is well presented;
rigorous investigation has been performed to a high technical & ethical standard;
the methods are described with sufficient detail & information to replicate.

Validity of the findings

The findings are well verified;
all underlying data have been provided;
the conclusions are well stated.

Additional comments

After revision, this paper is well written and all my concerns have been properly addressed. I think this paper deserves to be published in the current form.

Reviewer 2 ·

Basic reporting

The manuscript has now been thoroughly improved and I am satisfied with the revisions.

Experimental design

It's fine

Validity of the findings

It's fine

Additional comments

Thanks for your extensive changes and I think the article is good now. I still recommend that you go through to double check any grammar or editorial issues.